

# Detection and characterization of ESBL-producing *Escherichia coli* and additional co-existence with *mcr* genes from river water in northern Thailand

Kamonnaree Chotinantakul[1], Pattranuch Chusri[1] and Seiji Okada[2,3]

[1] School of Medicine, Mae Fah Luang University, Chiang Rai, Thailand
[2] Division of Hematopoiesis, Joint Research Center for Human Retrovirus Infection, Kumamoto University, Kumamoto, Kumamoto, Japan
[3] Graduate School of Medical Sciences, Kumamoto University, Kumamoto, Kumamoto, Japan

## ABSTRACT

**Background:** Extended-spectrum β-lactamase producing *Escherichia coli* (ESBL-producing *E. coli*) have emerged, causing human and animal infections worldwide. This study was conducted to investigate the prevalence and molecular genetic features of ESBL-producing and multidrug-resistant (MDR) *E. coli* in river water.

**Methods:** A total of 172 *E. coli* samples were collected from the Kok River and Kham River in Chiang Rai, Thailand, during a 10-month period (2020–2021).

**Results:** We detected 45.3% of *E. coli* to be MDR. The prevalence of ESBL-producers was 22%. Among those ESBL-producing strains, CTX-M-15 (44.7%) was predominantly found, followed by CTX-M-55 (26.3%), CTX-M-14 (18.4%), and CTX-M-27 (10.5%). The $bla_{TEM-1}$ and $bla_{TEM-116}$ genes were found to be co-harbored with the $bla_{CTX-M}$ genes. Mobile elements, *i.e.*, IS*Ecp*1 and Tn*3*, were observed. Twelve plasmid replicons were found, predominantly being IncF (76.3%) and IncFIB (52.6%). Whole genome sequencing of ten selected isolates revealed the co-existence of ESBL with *mcr* genes in two ESBL-producing *E. coli*. A wide diversity of MLST classifications was observed. An *mcr-1.1-pap2* gene cassette was found to disrupt the PUF2806 domain-containing gene, while an *mcr-3.4* contig on another isolate contained the *nimC/nimA-mcr-3.4-dgkA* core segment.

**Discussion:** In conclusion, our data provides compelling evidence of MDR and ESBL-producing *E. coli*, co-existing with *mcr* genes in river water in northern Thailand, which may be disseminated into other environments and so cause increased risks to public health.

# INTRODUCTION

*Escherichia coli* (*E. coli*) are commensal bacteria in humans and animals. However, *E. coli* is a commonly implicated bacteria, that can cause a variety of diseases, including diarrhea, septicemia, and urinary tract infection. Because *E. coli* can acquire antimicrobial resistance

Corresponding authors
Kamonnaree Chotinantakul, kamonnaree.cho@mfu.ac.th
Seiji Okada, okadas@kumamoto-u.ac.jp

genes *via* horizontal gene transfer, multidrug-resistant *E. coli* have been extensively found (*Razavi et al., 2020*). These infections are frequently associated with high morbidity and mortality in affected patients. The presence of *E. coli* expressing extended-spectrum β-lactamases (ESBLs) activity in patients, healthy carriers, and the environment has been reported in Thailand, (*Runcharoen et al., 2017*; *Saekhow & Sriphannam, 2021*; *Thamlikitkul, Tangkoskul & Seenama, 2019*).

The ESBLs phenotype, which can be produced by gram-negative bacteria, mediates the resistance to third-generation cephalosporins and monobactams. CTX-M has emerged as the most common ESBL type, displacing TEM-1 and -2 and SHV-1 (*Ruppé, Woerther & Barbier, 2015*). CTX-M enzymes are composed of five groups, which groups 1, 9, and 2 are commonly found in hospital settings and communities (*Bonnet, 2004*).

Regarding epidemiological studies of ESBL-producing *E. coli*, tracking the route and spreading in different environments, several studies focused on its presence, particularly in pigs and chickens (*Lay et al., 2021*; *Nahar et al., 2018*; *Seenama, Thamlikitkul & Ratthawongjirakul, 2019*). The widespread misuse of antibiotics in farming and pork meat was also studied (*Tansawai et al., 2018*). It has been found that contamination in farm wastewater could also occur (*Saekhow & Sriphannam, 2021*). The study of contamination of ESBL-producing *E. coli* in cultivated soils demonstrated that they could survive for extended periods of time (*Hartmann et al., 2012*). Outbreaks due to surface water contamination in association with extreme precipitation were implicated as a public health concern (*Curriero et al., 2001*). Thus, water could be the source of dissemination of ESBL-producing *E. coli* over extensive areas, including water sources for human drinking water (*Mahmud et al., 2020*). Several reports showing the presence of antibiotic-resistant *E. coli*, including ESBL isolates from water environments, have been published in other countries (*Banu et al., 2021*; *Hassen et al., 2020*; *Murugadas et al., 2021*). Nonetheless, the epidemiological data available for the contamination of ESBL-producing *E. coli* in water rivers is still limited in Thailand.

The Kok River, which has its source in Myanmar and flows through Chang Rai and Chiang Mai provinces in northern Thailand, is a 285 km tributary river (leading to the larger Mekong River). Most of its length in Thailand is in Chiang Rai province, where it receives inputs from urban catchments in Mueang Chiang Ria district. The Kham River originates in Chiang Rai province and flows through to the Mekong River (85 km). Both rivers are used in agriculture, especially rice and various crops; this is the major land use in Chiang Rai (*Chantima et al., 2020*). Hence, the Kok and Kham Rivers provide the site for an epidemiological study of the multidrug-resistant (MDR) *E. coli* as this relates to the main use of water resources for people in many activities in Chiang Rai and may be the source of water-borne diseases. Therefore, this study aimed to determine the prevalence of MDR and ESBL-producing *E. coli* in river water. Furthermore, plasmid profiling and resistant genes were also characterized to clarify the possibility and extent of dissemination.
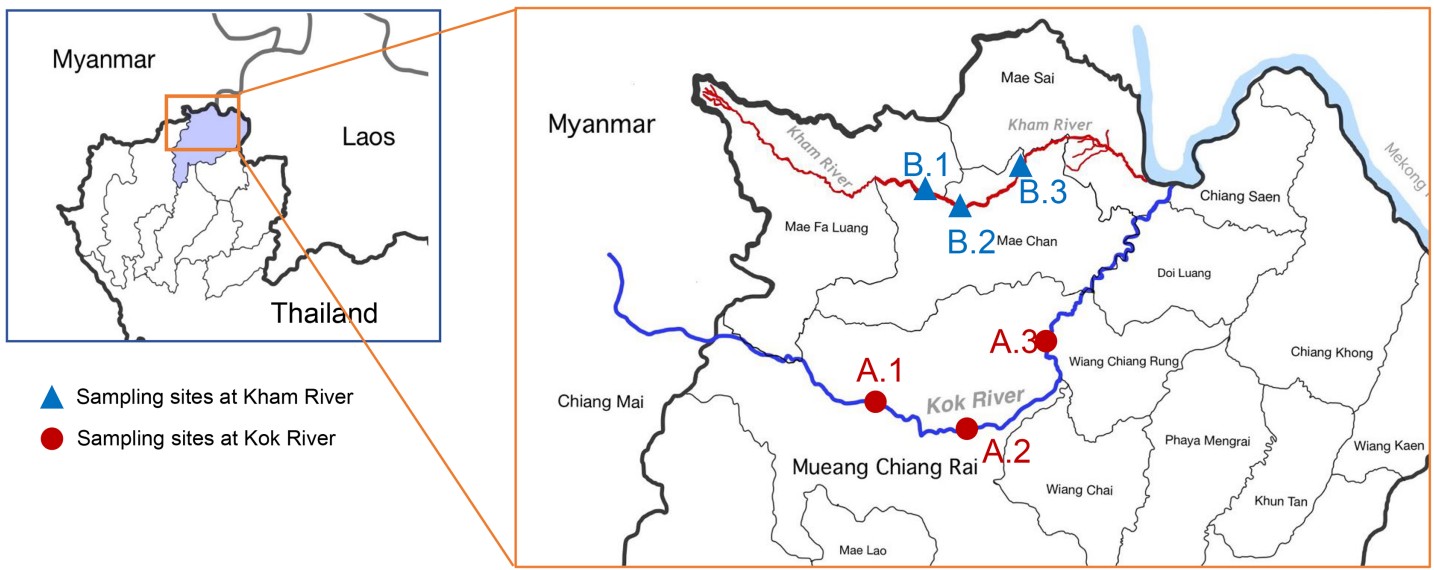

**Figure 1** **Map of the location of the sampling sites at the KokRiver and Kham River in Chiang Rai province.** A small box indicates the location of Chiang Rai province in northern Thailand (highlight area). Samples were collected from three sampling sites on each river. The blue line shows the Mae KokRiver whereas the red line represents the Kham River. The KokRiver passes through the Mueang Chiang Rai district, where site A.1 is upstream of the city, site A.2 is located at the center of the city, and site A.3 is downstream of the main city. The Kham River passes through the Mae Chan district, where site B.1 is upstream of the city, site B.2 is inside the city, and site B.3 is downstream of the city.

# MATERIALS AND METHODS

## Study area and sample collection

Water samples were collected from the two main rivers in Chiang Rai, Thailand (Kok River and Kham River). Sampling was performed on three sites at each river (Fig. 1). Site A.1 was located close to agricultural areas upstream of the Mueang Chiang Rai district. Site A.2 was located on the route of river flow close to the center of Chiang Rai city. Site A.3 was located downstream of the Kok River, after its passage through the main city to the urban areas with agricultural activity taking place alongside the river. For sampling at the Kham River, site B.1 was located near the transition to the agricultural areas above Mae Chan district. Site B.2 was located close to the community areas of the residents of the Mae Kham sub-district, where both urban and agricultural activities were taking place. Site B.3 was located downstream, being more agricultural in nature. The study design and field experiments were approved by the Research Council of Mae Fah Luang University (project number: 641C08004).

Between December 2021 and September 2022, water samples were obtained from sites A.1 to A.3 and sites B.1 to B.3 monthly. Water samples were collected at a depth of 30 cm below the surface of water with sterile bottles (500 ml/bottle) in triplicate at each sampling site. During the transportation, all samples were kept on ice and processed within 6 h of collection.

## Bacterial enumeration, *E. coli* identification and DNA isolation

Water samples were processed as described previously (*Purohit et al., 2020*). Briefly, 10-fold serial dilutions of water samples (10 ml) were prepared in sterile 0.9% normal saline and processed by standard membrane filtration technique using 47 mm in diameter and a pore size of 0.45 µm membrane filters (Merck Millipore, Darmstadt, Germany). After that, the membranes were placed on Coliform agar (Merck Millipore, Darmstadt, Germany) for 24 h at 37 °C for cultivation and manual counting of colonies. The total coliform count was enumerated in colony-forming units (CFUs)/ml. Three independent assays were performed for each sampling site, and technical triplicates were used. For the selection of ESBL-producing *E. coli*, water samples were inoculated on the selective medium CHROMagar ESBL (dark pink-red colony; CHROMagar, Paris, France) for 24 h at 37 °C. The identification of 6–10 *E. coli* isolates per water-river sampling site was followed by biochemical tests (Indole, motile, citrate, methyl red, and Voges-Proskauer test) and PCR amplification of *yaiO* and *uidA* genes (*Molina et al., 2015*). Genomic DNA was extracted using the boiling method, while plasmid DNA was extracted using the Nucleospin plasmid extraction kit (Macherey-Nagel, Duren, Germany). The DNA was stored at −20 °C and subjected to a PCR-based assay.

## Antimicrobial susceptibility testing

The confirmed *E. coli* colonies were subjected to antibiotic susceptibility testing with eight commonly used classes of antibiotics by the Kirby Bauer disc diffusion test on Muller Hinton Agar (Himedia, Mumbai, India). The antimicrobials selected were ciprofloxacin, nalidixic acid, chloramphenicol, streptomycin, gentamicin, meropenem, ertapenem, tetracycline, amoxicillin-clavulanic acid, ampicillin, trimethoprim/sulfamethoxazole, cefoxitin, cefepime, ceftazidime, and cefotaxime (Oxoid, Hampshire, UK). The procedure and interpretation were performed according to the Clinical and Laboratory Standard Institute guidelines (*CLSI, 2020*). Intermediate results were categorized as resistant. Multidrug resistance (MDR) bacteria were defined by resistance to at least one agent in three or more antimicrobial classes. For quality control, the *E. coli* reference strain ATCC 25922 was used. ESBL-producing strains were confirmed by the combination disc diffusion test, where an increase in the inhibition zone diameter of 5 mm for a combination disc *vs.* either ceftazidime or ceftriaxone confirmed ESBL production. A CLSI broth microdilution was used to determine the MIC of colistin in isolates harboring *mcr* genes.

## Plasmid replicon typing

Plasmid typing was characterized by five multiplex (M)-PCR, including multiplex 1 for HI1, HI2 and I1-Iγ, multiplex 2 for X, L/M and N, multiplex 3 for FIA, FIB and W, multiplex 4 for Y, P and FIC, and multiplex 5 for A/C, T and FIIAs. Three simplex PCRs were used to detected for F, K, and B/O (*Carattoli et al., 2005*).

## Phylogenetic typing, β-lactamase gene, integrons and mobile genetic elements

Phylogenetic groups of *E. coli* (A, B1, B2, C, D, E, F, and *Escherichia* cryptic clade I) were classified by PCR as described previously (*Clermont et al., 2013*). The $bla_{CTX-M}$ (group 1, 2, and 9) genes were detected *via* M-PCRs (*Dallenne et al., 2010*), and the $bla_{TEM}$ and $bla_{SHV}$ genes were detected using a simplex PCR (*Pitout et al., 1998*). The presence of integrons (*intI1* and *int2*) (*Kurekci et al., 2017*), transposon (Tn*3*) (*Gregova, Kmet & Szaboova, 2021*), and insertion sequence (IS*Ecp1*) (*Eckert, Gautier & Arlet, 2006*) were detected by PCR as described previously. DNA sequencing of PCR products (CTX-M1, M9, and TEM) was used.

## Whole genome sequencing and analysis

We selected ten ESBL-producing *E. coli* isolates from both the Kok River and the Kham River for whole-genome sequencing, as these isolates demonstrated wider antimicrobial resistance patterns with different classes of antibiotics. Whole-genome sequencing was performed by Macrogen (Seoul, South Korea). AxyPrep Bacterial Genomic DNA (Axygen Biosciences, Hangzhou, China) was used to perform DNA extraction from an overnight culture of all selected *E. coli* isolates. The sequencing DNA library was prepared using the TruSeq Nano DNA Kit (Illumina, San Diego, CA, USA). Whole genome sequencing was performed on the Illumina HiSeq 2,500 with 101 bp paired-end reads. The average number of assembled contigs per sample was 127 (range 67 to 220), the average N50 was 185 kb (range 93 kb to 242 kb), and the total assembly length was 4.7 to 5.4 megabases (Mb) (Table S1). Raw read quality was checked using FASTQC software (*Wingett & Andrews, 2018*) and the adaptors and poor-quality reads were removed by using Fastp (*Chen et al., 2018*). Genome assemblies were performed using Unicycler (*Wick et al., 2017*) and annotated with Prokka (*Seemann, 2014*) at default settings. Genome assemblies were evaluated for quality by Quast (*Gurevich et al., 2013*). Antimicrobial resistance genes were identified by ABRicate, which included the databases of Resfinder (*Zankari et al., 2012*), PointFinder (*Zankari et al., 2012*), CARD (*Alcock et al., 2020*), PlasmidFinder (*Carattoli et al., 2014*), and SerotypeFinder (*Joensen et al., 2015*). All gene predictions were called by applying a select threshold for identification and a minimum length of 95% and 80%, respectively. For sequence type analysis, raw data generated from the Illumina platform were submitted to Enterobase (https://enterobase.warwick.ac.uk/), and the multilocus sequence typing (MLST) was determined with MLST 2.0 (*Larsen et al., 2012*). A circular comparison between the genome assembly contig carrying *mcr-1*, with the highest similarity reference plasmids from the NCBI database, was generated using the circular genome viewer (CGView) tool, freely available at Proksee website (https://proksee.ca) (*Stothard, Grant & Van Domselaar, 2019*). The whole-genome sequence MLST (wgMLST) and canonical wgMLST (cano-wgMLST) analyses of selected 10 ESBL-producing *E. coli* isolates were conducted using cano-wgMLST_BacCompare web-based tool (*Liu, Lin & Chen, 2019*). The whole-genome sequences were compared with the constructed pan-genome allele database (PGAdb), using BLASTN v2.2.30+ program (alignment coverage ≥90%; alignment identity ≥90%). The wgMLST tree was constructed using the

whole-genome scheme, while the cano-wgMLST tree was built using the highly discriminatory loci among isolates. The dendrogram of both trees were visualized with iTOL v6 (http://itol.embl.de) (Ciccarelli et al., 2006).

The draft genomes of all 10 ESBL-producing *E. coli* strains in this work has been deposited under the BioProject accession number PRJNA846957 with BioSample accessions: SAMN28906491–SAMN28906500.

### Statistical analysis

An Unpaired *t*-test was applied to compare the means of the CFU/ml between each site of the river water collected in each month ($p < 0.05$). Independent *t*-test was performed where the sample mean values were normally distributed. Three replicates (independent experiments) were performed for all assays. Descriptive statistical parameters, such as the mean and standard deviation, were applied to the data.

## RESULTS

### Distribution of coliform bacteria

The level of coliform bacteria CFU/ml for the three sites of the Kok River was between $14.78 \times 10^3$ and $109.00 \times 10^3$ (mean $54.23 \pm 23.31 \times 10^3$) and the Kham River was between $6.56 \times 10^3$ and $137.33 \times 10^3$ (mean $59.08 \pm 35.54 \times 10^3$). Overall, the number of coliform bacteria peaked in June (mean total $96.52 \pm 8.85 \times 10^3$ CFU/ml) and August (mean total $123.37 \pm 10.85 \times 10^3$ CFU/ml) for the Kok River and Kham River, respectively, this being the rainy season. Generally, the number of coliform bacteria was not different for each sampling site at Kok River (Fig. 2A). In January, the colony count at site A.1 ($30.89 \times 10^3$ CFU/ml) was lower than at site A.3 ($57.33 \times 10^3$ CFU/ml, $p < 0.01$), while in February, site A.1 ($37.33 \times 10^3$ CFU/ml) was lower than site A.2 ($86 \times 10^3$ CFU/ml, $p < 0.01$). On the other hand, in March, the colony count at site A.1 ($53.44 \times 10^3$ CFU/ml) was higher than at site A.3 ($37.89 \times 10^3$ CFU/ml, $p = 0.03$). Moreover, in September, CFU/ml of coliform bacteria were higher at site A.1 ($91.67 \times 10^3$ CFU/ml) than at sites A.2 ($79.33 \times 10^3$ CFU/ml, $p < 0.01$) and A.3 ($58.56 \times 10^3$ CFU/ml, $p < 0.01$).

In December, January, March, and April, the colony count at sites B.1 of the Kham River ($6.89 \times 10^3$, $6.56 \times 10^3$, $7.22 \times 10^3$, and $9.22 \times 10^3$ CFU/ml, respectively) (Fig. 2B) had a lower number than sites B.2 ($21.11 \times 10^3$ ($p < 0.001$), $31.11 \times 10^3$ ($p < 0.01$), $29.33 \times 10^3$ ($p < 0.01$), and $35.78 \times 10^3$ ($p < 0.001$) CFU/ml, respectively) and B.3 ($23.78 \times 10^3$ ($p < 0.001$), $88.56 \times 10^3$ ($p < 0.001$), $28.33 \times 10^3$ ($p < 0.01$), and $58.33 \times 10^3$ ($p < 0.001$) CFU/ml, respectively) (Fig. 2B). In addition, in July and August, the colony count at sites B.1 ($74.44 \times 10^3$ and $110.89 \times 10^3$ CFU/ml, respectively) had a lower number than sites B.3 ($111.33 \times 10^3$ ($p < 0.01$) and $137.33 \times 10^3$ ($p = 0.02$) CFU/ml, respectively). In June, however, the CFU/ml observed at site B.2 ($79.56 \times 10^3$ CFU/ml) was lower than at site B.1 ($82.44 \times 10^3$ CFU/ml, $p = 0.01$).

### *E. coli* and antibiotic susceptibility test

A total of 172 *E. coli* isolates were collected, including 74 isolates from the Kok River and 98 isolates from the Kham River. Of the *E. coli* isolates obtained from both rivers,

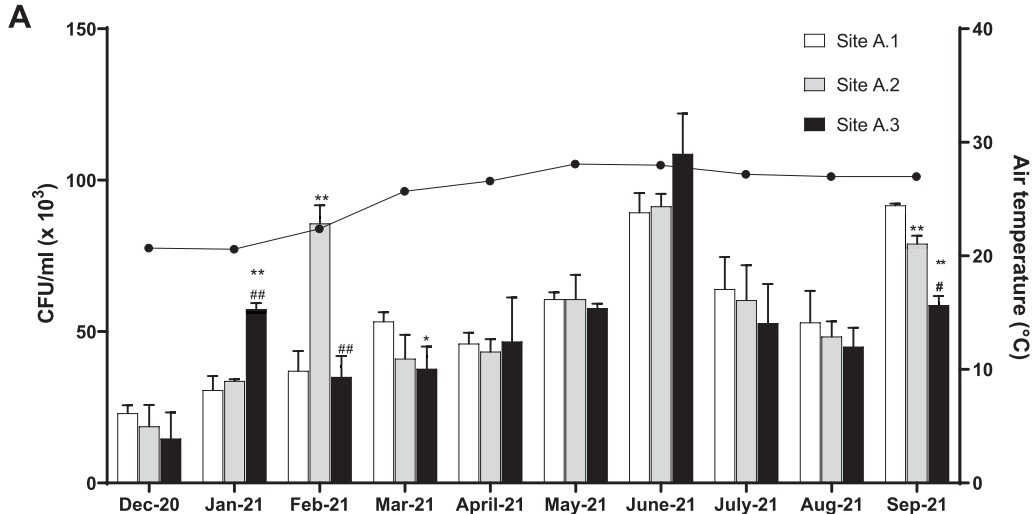

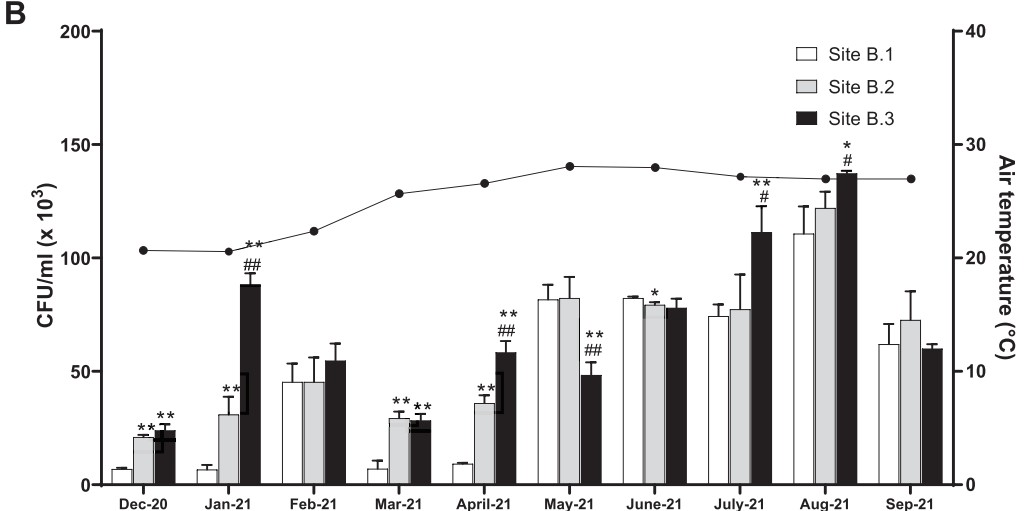

**Figure 2 Total coliform bacteria in the (A) KokRiver and (B) Kham River in northern Thailand.** CFUs were counted from each collecting site. The columns represent the mean plus or minus standard deviation of three independent experiments, with triplicates. Statistical differences were analyzed with an unpaired $t$-test. Values that are significantly different are indicated by asterisks as follows: $^*p < 0.05$, $^{**}p < 0.01$ when compared to site 1; $^\#p < 0.05$, $^{\#\#}p < 0.01$ when compared to site 2.

45.3% (78/172) were positive for MDR. Most isolates from the two rivers were resistant to ampicillin (71.5%, 123/172), followed by tetracycline (46.5%, 80/172), streptomycin (32.6%, 56/172), amoxicillin-clavulanic acid (29.7%, 51/172), ciprofloxacin (26.7%, 46/172), cefotaxime (25.6%, 44/172) and cefepime (24.4%, 42/172). A few isolates were resistant to nalidixic acid (20.9%, 36/172), trimethoprim/sulfamethoxazole (19.2%, 33/172), ceftazidime (18%, 31/172), chloramphenicol (16.3%, 28/172), gentamicin (9.9%, 17/172), meropenem (2.3%, 4/172), cefoxitin (1.7%, 3/172), and ertapenem (0.6%, 1/172). No pan-drug resistance was observed. The percentage of antibiotic resistant *E. coli* from each river is shown in Fig. 3. Overall, a total of 80 antibiogram profiles were obtained

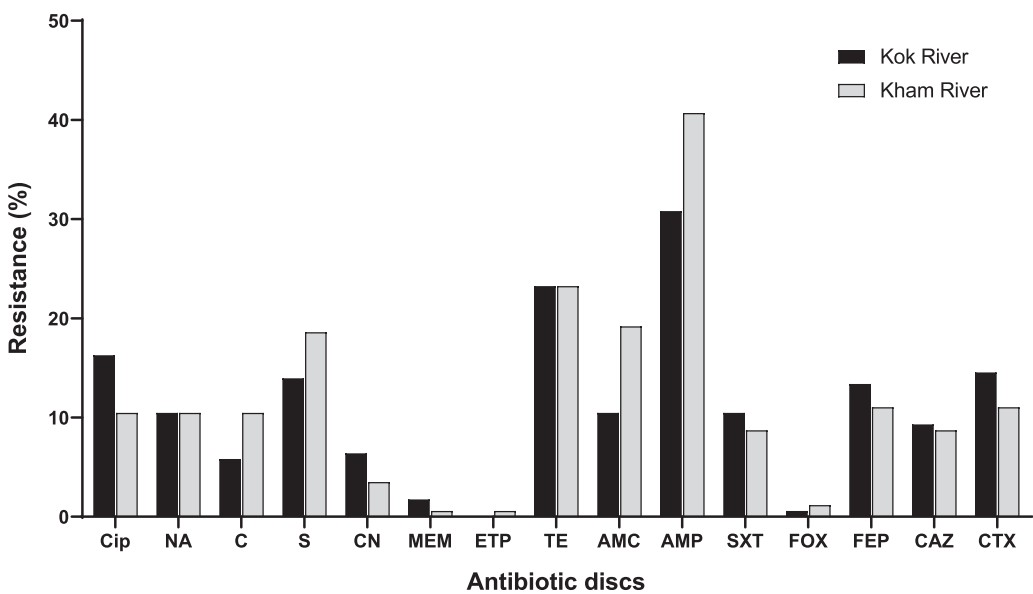

**Figure 3 Antibiotic resistance of *E. coli* from the KokRiver and Kham River in northern Thailand.** Cip, ciprofloxacin; NA, nalidixic acid; C, chloramphenicol; S, streptomycin; CN, gentamicin; MEM, meropenem; ETP, ertapenem; TE, tetracycline; AMC, amoxicillin-clavulanic acid; AMP, ampicillin; STX, trimethoprim/sulfamethoxazole; FOX, cefoxitin; FEP, cefepime; CAZ, ceftazidime; CTX, cefotaxime.

(Table S2). Furthermore, a total of 22.1% (38/172) of ESBL-producing *E. coli* isolates were collected. ESBL-producing *E. coli* isolates from the Kok River and Kham River were 31.1% (23/74) and 15.3% (15/98), respectively. Of note, all ESBL isolates were sensitive to ertapenem but resistant to ampicillin. Resistance to ciprofloxacin, tetracycline, and streptomycin was the most common trait (Fig. 3).

## Phylogenetic grouping

Phylogenetic typing revealed that phylogroups B1, A, and C were the predominant types and were detected in 46.5% (80/172), 17.4% (30/172), and 16.3% (28/172), respectively. Other phylogroups were found at a lower frequency, including phylogroups E (8.7%, 15/172), B2 (4.7%, 8/172), D (4.1%, 7/172), and F (2.3%, 4/172).

## Characterization of β-lactamase gene and genetic elements

All 38 ESBL isolates contained $bla_{CTX-M}$, consisting of $bla_{CTX-M-15}$ (44.7%, 17/38), $bla_{CTX-M-55}$ (26.3%, 10/38), $bla_{CTX-M-14}$ (18.4%, 7/38), and $bla_{CTX-M-27}$ (10.5%, 4/38). $bla_{TEM-1}$ and $bla_{TEM-116}$ genes were co-harbored with the $bla_{CTX-M}$ gene in 23.7% (9/38) and 2.6% (1/38), respectively, whereas $bla_{SHV}$ was not detected. The presence of integrase genes was found to be 55.3% of *Int*1 genes (21/38) and 5.3% of *Int*2 genes (2/38), and one isolate contained both *Int*1 and *Int*2 genes. IS*Ecp1* and Tn*3* genes were found at 55.3% (21/38) and 21.1% (8/38), respectively (Table 1).

**Table 1 Characteristics of ESBL-producing *E. coli*.**

| Strains | Month | Site | Phylogroup[a] | Beta-actamases[b] | Integrons/ transposons[c] | Plasmid Replicon[d] | Antibiotic resistance patterns[e] |
|---|---|---|---|---|---|---|---|
| EH2101 | Dec-20 | Kok River | A | CTX-M-15 | IS*Ecp1* | F, FIB, I1-Iγ, K, B/O | 40 |
| EH2102 | Dec-20 | Kok River | B2 | CTX-M-15 | *Int1* | F, FIA, I1-Iγ | 11 |
| EH3101 | Dec-20 | Kok River | D | CTX-M-15 | IS*Ecp1* | F | 24 |
| EH1201 | Jan-21 | Kok River | B1 | CTX-M-14 | IS*Ecp1* | F, FIB, I1-Iγ, FIIAs, X | 13 |
| EH1203 | Jan-21 | Kok River | B1 | CTX-M-15 | - | F, FIB, I1-Iγ, Y, K, B/O | 59 |
| EH2201 | Jan-21 | Kok River | B2 | CTX-M-27 | *Int1* | F, FIA, I1-Iγ | 29 |
| EH2204 | Jan-21 | Kok River | A | CTX-M-14 | - | F, HI1, K | 63 |
| EH3201 | Jan-21 | Kok River | B1 | CTX-M-15 | - | F, I1-Iγ, K, B/O | 50 |
| EH1303 | Feb-21 | Kok River | B1 | CTX-M-15 | IS*Ecp1* | F, FIA, Y, K | 49 |
| EH1306 | Feb-21 | Kok River | B1 | CTX-M-14 | *Int1*, IS*Ecp1* | F, FIB, HI2, K | 21 |
| EH1307 | Feb-21 | Kok River | C | CTX-M-14 | *Tn3*, IS*Ecp1* | F, FIB, I1-Iγ, A/C, B/O, FIIAs | 13 |
| EH2301 | Feb-21 | Kok River | A | CTX-M-55, TEM-1 | *Int1*, *Tn3*, IS*Ecp1* | F, HI2, X | 7 |
| EH3301 | Feb-21 | Kok River | A | CTX-M-15 | - | F, Y, K, B/O | 51 |
| EH3302 | Feb-21 | Kok River | A | CTX-M-14 | *Tn3* | F, FIB, Y | 39 |
| EH1401 | Mar-21 | Kok River | B1 | CTX-M-15, TEM-1 | *Int1*, *Tn3*, IS*Ecp1* | – | 18 |
| EH1402 | Mar-21 | Kok River | A | CTX-M-55 | *Int1* | F, FIB, HI1 | 1 |
| H2404 | Mar-21 | Kok River | B1 | CTX-M-15 | IS*Ecp1* | F, FIB, HI1, I1-Iγ | 32 |
| EH2401 | Mar-21 | Kok River | B2 | CTX-M-15 | *Int1* | F, FIA, I1-Iγ | 8 |
| EH2402 | Mar-21 | Kok River | B1 | CTX-M-55, TEM-1 | *Int1*, *Tn3*, IS*Ecp1* | F, FIB, I1-Iγ | 4 |
| EH6303 | May-21 | Kok River | D | CTX-M-15 | - | F, I1-Iγ, B/O | 58 |
| EH8203 | Jul-21 | Kok River | D | CTX-M-15, TEM-1 | *Int1*, *Int2*, IS*Ecp1* | F, HI2, K | 2 |
| EH9101 | Aug-21 | Kok River | D | CTX-M-15, TEM-1 | *Int1*, *Int2*, IS*Ecp1* | F, FIB | 5 |
| EH9102 | Aug-21 | Kok River | C | CTX-M-27 | *Int1* | FIA, FIB | 15 |
| EK3101 | Dec-20 | Kham River | B1 | CTX-M-55, TEM-116 | *Int1*, IS*Ecp1* | Y | 10 |
| EK1201 | Jan-21 | Kham River | B1 | CTX-M-55 | *Int1*, IS*Ecp1* | Y | 3 |
| EK1301 | Feb-21 | Kham River | B1 | CTX-M-15 | IS*Ecp1* | FIA | 34 |
| EK1302 | Feb-21 | Kham River | F | CTX-M-55, TEM-1 | *Int1*, IS*Ecp1*, *Tn3* | F, FIB, I1-Iγ | 14 |
| EK2302 | Feb-21 | Kham River | B1 | CTX-M-27 | - | F, FIB, I1-Iγ | 57 |
| EK2303 | Feb-21 | Kham River | B1 | CTX-M-14, TEM-1 | *Int1*, IS*Ecp1*, *Tn3* | F, FIA, FIB, Y, K | 27 |
| EK3304 | Feb-21 | Kham River | E | CTX-M-55 | - | F, FIB | 36 |
| EK3305 | Feb-21 | Kham River | B1 | CTX-M-55 | IS*Ecp1* | F, FIB | 37 |
| EK1401 | Mar-21 | Kham River | B1 | CTX-M-55 | *Int1* | F, FIB, HI2 | 19 |
| K1506 | Apr-21 | Kham River | C | CTX-M-14 | *Int1* | Y | 47 |
| EK2501 | Apr-21 | Kham River | D | CTX-M-15, TEM-1 | *Int1*, IS*Ecp1*, *Tn3* | Y | 15 |
| EK2503 | Apr-21 | Kham River | E | CTX-M-27 | *Int1* | F, FIA, FIB | 15 |
| EK2504 | Apr-21 | Kham River | D | CTX-M-15, TEM-1 | *Int1*, IS*Ecp1*, *Tn3* | Y | 15 |
| EK8301 | Jul-21 | Kham River | B1 | CTX-M-15 | IS*Ecp1* | FIIAs | 45 |
| EK9101 | Aug-21 | Kham River | B1 | CTX-M-55 | *Int1* | F, FIB, HI2, Y | 17 |

**Notes:**
[a] Phylogroup characterized by clement typing.
[b] Beta-lactamase.
[c] Integrons/transposon.
[d] Plasmid replicon were determined using PCR.
[e] Antibiotic resistance patterns are shown in Table S2.

## Plasmid replicon typing

In total, twelve plasmid replicons were detected in the present work. The predominant types were F, FIB, I1-Iγ, Y, and K, which were detected in 76.3% (29/38), 52.6% (20/38), 34.2% (13/38), 34.2% (13/38), and 26.3% (10/38), respectively. Plasmid replicon types L/M, N, P, T, and W were not detected in this study. Other replicons were found with low prevalence, including FIA (18.4%, 7/38), B/O (15.8%, 6/38), HI2 (13.2%, 5/38), FIIAs (7.9%, 3/38), HI1 (7.9%, 3/38), X (2.0%, 2/38), A/C (2.6%, 1/38), and FIC (2.6%, 1/38).

## Whole genome sequencing

Ten ESBL-producing *E. coli* were selected for whole genome sequencing (WGS) analysis to identify the genes and plasmid types that are responsible for resistance. All 10 ESBL-producing *E. coli* contained more than five different types of acquired resistance genes as well as at least one resistance plasmid (Table 2). The other antimicrobial resistance genes in the ESBL-producing *E. coli*, including aminoglycosides, fluoroquinolones, macrolides, chloramphenicol, polymyxin, sulfonamide, tetracycline, and trimethoprim, are shown in Table 2. Moreover, quinolone resistance was observed due to mutations in chromosomal genes *gyrA*(S83L, D87N), *parC*(S80I, E84K, E84V), and *parE*(S458A, I529L). Substitution at S83L and D87N in *gyrA* was predominant. The isolates EK2501, EK2504, and EK9101 did not contain quinolone resistance due to mutation.

Additionally, the co-occurrence of *mcr-1.1*, $bla_{TEM-1}$, and $bla_{CTX-M55}$ was found in the EH2301 isolate, while *mcr-3.4*, $bla_{TEM-1}$, and $bla_{CTX-M55}$ were both found in the EK9101 isolate (Table 2). These two isolates exhibited phenotypic resistance to colistin, by broth microdilution, showing that the minimal inhibitory concentration (MIC) values of these *mcr*-harboring isolates were 4 μg/ml (the MIC value of ≥4 μg/ml confirmed resistance according to the 2020 CLSI M100-30 guidelines). The *mcr-1.1* gene in the EH2301 isolate was in a contig that is presumed to be part of an IncX4 plasmid. The *mcr-1.1* harboring IncX4 contig in EH2301 (contig 45; 32,787 bp in length) was similar to the *mcr-1* harboring IncX4 plasmids isolated from pig origin in Thailand (pCP52E-IncX4, accession number CP075733), duck origin in Thailand (PN23, accession number MG557852), and human origin in China (pEC931_mcr, accession number CP049122) (Fig. 4.). When we compared the contig 45 to the reference plasmid sequences, 99% nucleotide identity was observed. The aligned sequence was covered throughout the contig 45. However, the *mcr-3.4* gene could not be predicted on the contig because the plasmid marker was not observed by PlasmidFinder 2.0.1. Moreover, when we aligned the contig 113, carrying the *mcr-3.4* gene in the NCBI database, it was shown to be on the plasmid region of pK18EC051 (Accession number CP049300; 99% identity). The genetic organization of the *mcr* genes in these isolates is outlined in Fig. 5. The genomic context of the *mcr-1.1-pap2* cassette in EH2301 disrupted a pre-existing DUF2806-demain containing gene and contained the flanking upstream and downstream regions with a DUF2726 domain-containing gene and pseudo-methyltransferase, respectively. The genomic cassette demonstrated 100% nucleotide identity (BLAST aligned with accession number CP063335). The upstream and downstream genetic organization of the *mcr-3.4* gene was different from that of the *mcr-*

**Table 2 Molecular characteristics of 10 ESBL-producing *E. coli*.**

| Isolate | MLST[a] | Serotypes[b] | Resistance genes[c] | | | | | | | | | Plasmids[d] |
| --- | --- | --- | --- | --- | --- | --- | --- | --- | --- | --- | --- | --- |
| | | | Aminoglycoside | Beta-lactam | Quinolone resistance gene/point mutation | Macrolide, Lincosamind, Streptogramin B | Phenicol | Polymyxin | Sulfonamide | Tetracycline | Trimethoprim | |
| EH1201 | 224 | O8:H23 | *aac(3)-IId*, *ant(3")-Ia* | *bla*CTX-M-14 | *gyrA*(S83L, D87N), *parC*(S80I), *parE*(S458A) | *mdf(A)*, *erm(42)*, *erm(B)*, *mph(A)* | *floR* | | *sul2* | *tet(X)* | | IncFIC(FII), IncX1, IncFIB(AP001918), Col(MG828) |
| EH1307 | 13160 | -:H4 | *aac(3)-IId*, *aph(3')-Ib*, *aph(6)-Id* | *bla*CTX-M-14 | *qnrS1*, *gyrA*(S83L) | *mdf(A)* | *floR* | | *sul2* | *tet(A)* | | p0111, IncFIC(FII), IncA/C2, IncB/O/K/Z, IncFIB(AP001918), ColpVC, Col(MG828) |
| EH2102 | 131 | O25:H4 | *aadA5* | *bla*CTX-M-15, *bla*OXA-1 | *aac(6')-Ib-cr*, *gyrA*(S83L, D87N), *parC*(S80I, E84V), *parE*(I529L) | *mdf(A)*, *mph(A)* | | | *sul1* | *tet(A)* | *dfrA17* | IncFII, IncFIA, Col156, Col(BS512), Col(MG828) |
| EH2301 | 1421 | O9:H4 | *aac(3)-IId*, *aadA2*, *ant(3")-Ia*, *aph(3')-Ib* | *bla*CTX-M-55, *bla*TEM-1B | *qnrS1*, *gyrA*(S83L), *parC*(S80I) | *mdf(A)*, *lnu(F)* | *cmlA1*, *floR* | *mcr-1.1* | *sul2, sul3* | *tet(A), tet(X)* | *dfrA12* | IncR IncX1, IncX4, |
| EH9101 | 648 | O102:H6 | *aac(3)-IIa*, *aph(6)-Id* | *bla*CTX-M-15, *bla*OXA-1, *bla*TEM-1B | *aac(6')-Ib-cr*, *gyrA*(S83L, D87N), *parC*(S80I), *parE*(S458A) | *mdf(A)* | | | *sul2* | *tet(A)* | *dfrA14* | IncFII(pRSB107), IncFIB(AP001918), Col(BS512), Col(MG828) |
| EK1201 | 224 | O8:H23 | *aac(3)-IIa*, *aadA2*, *ant(3")-Ia*, *aph(3')-Ib*, *aph(6)-Id* | *bla*CTX-M-55 | *gyrA*(S83L, D87N), *parC*(S80I), *parE*(S458A) | *mdf(A)* | *cmlA1*, *floR* | | | *tet(A), tet(M)* | *dfrA12* | IncFIB(pHCM2), IncFIA(HI1), IncFIB(K), IncX1, Col440I |
| EK2303 | 603 | O175:H16 | *aac(3)-IId*, *aadA2*, *aph(3')-Ia*, *aph(3')-Ib*, *aph(6)-Id* | *bla*CTX-M-14, *bla*TEM-1B | *gyrA*(S83L, D87N), *parC*(E84K) | *mdf(A)*, *lnu(F)* | | | *sul2* | *tet(B)* | | IncFIA, IncFIB(AP001918), IncY, Col156, Col(MG828) |
| EK2501 | 69 | O17 or O44 or O77: H18 | *aph(3')-Ib*, *aph(6)-Id* | *bla*CTX-M-15, *bla*TEM-1B | *qnrS1* | *mdf(A)* | | | *sul2* | *tet(A)* | *dfrA14* | IncY |

(Continued)

## Table 2 (continued)

| Isolate | MLST[a] | Serotypes[b] | Resistance genes[c] | | | | | | | | | | Plasmids[d] |
|---------|---------|--------------|---------------------|---|---|---|---|---|---|---|---|---|-------------|
| | | | Aminoglycoside | Beta-lactam | Quinolone resistance gene/ point mutation | Macrolide, Lincosamind, Streptogramin B | Phenicol | Polymyxin | Sulfonamide | Tetracycline | Trimethoprim | | |
| EK2504 | 69 | O17 or O44 or O77: H18 | *aph(3″)-Ib*, *aph(6)-Id* | *bla*CTX-M-15, *bla*TEM-1B | *qnrS1* | *mdf(A)* | | | *sul2* | *tet(A)* | *dfrA14* | | IncY |
| EK9101 | 5218 | O3:H7 | *aac(3)-IId*, *aadA2*, *ant(3″)-Ia*, *aph(3″)-Ib*, *aph(6)-Id* | *bla*CTX-M-55 | *qnrS1* | *mdf(A)* | *catA2*, *cmlA1* | *mcr-3.4* | *sul2, sul3* | | | | P0111, IncFIB(AP001918), IncI1, IncHI2A, IncHI2 |

**Notes:**
[a] MLST determined by https://enterobase.warwick.ac.uk/.
[b] Serotype determined by https://cge.food.dtu.dk/services/SerotypeFinder/.
[c] The antibiotic resistance genes were search with ResFinder 4.1.
[d] The plasmid markers were identified by PlasmidFinder 2.1.

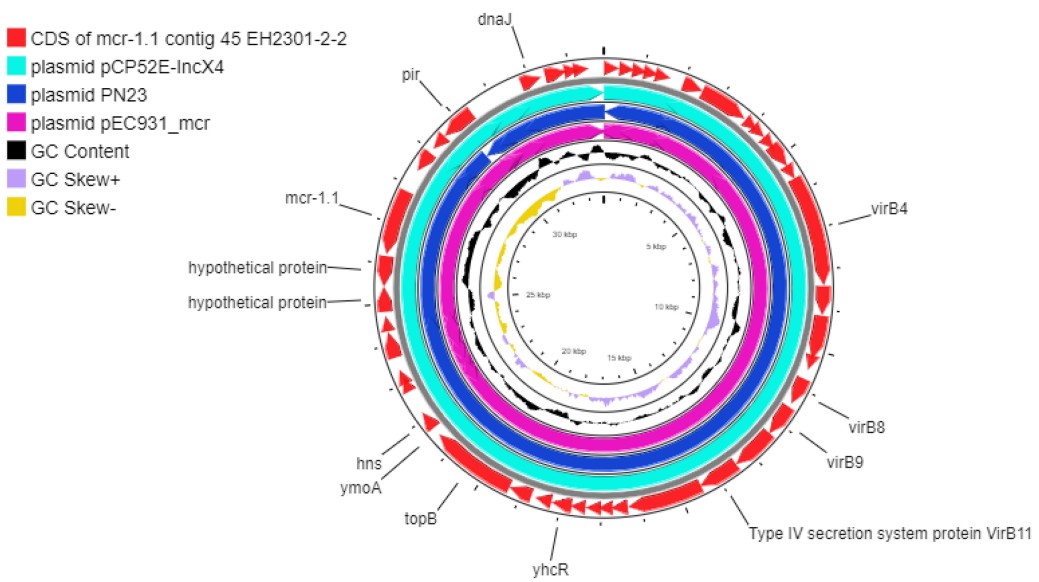

**Figure 4 Circular comparison between the *mcr-1.1*-carrying contig 45 from EH2301 to the most three identical IncX4 type plasmids carrying *mcr-1* deposited in the GenBank database.** The circles compared the genome sequences between the *mcr-1.1*-carrying contig 45, from EH2301, to three IncX4 type plasmids carrying *mcr-1* obtained from the GenBank database (accession numbers CP075733, MG557852, and CP049122) generated by the CGViewtool (https://proksee.ca). The outermost circle denotes the coding sequence of the contig 45 carrying *mcr-1.1* from the present work. The second to fourth circles represent the BLASTN comparison of the *mcr-1.1*-carrying IncX4 plasmid against the *mcr-1.1* harboring plasmids pCP52E-IncX4, PN23, and pEC931_mcr, respectively.

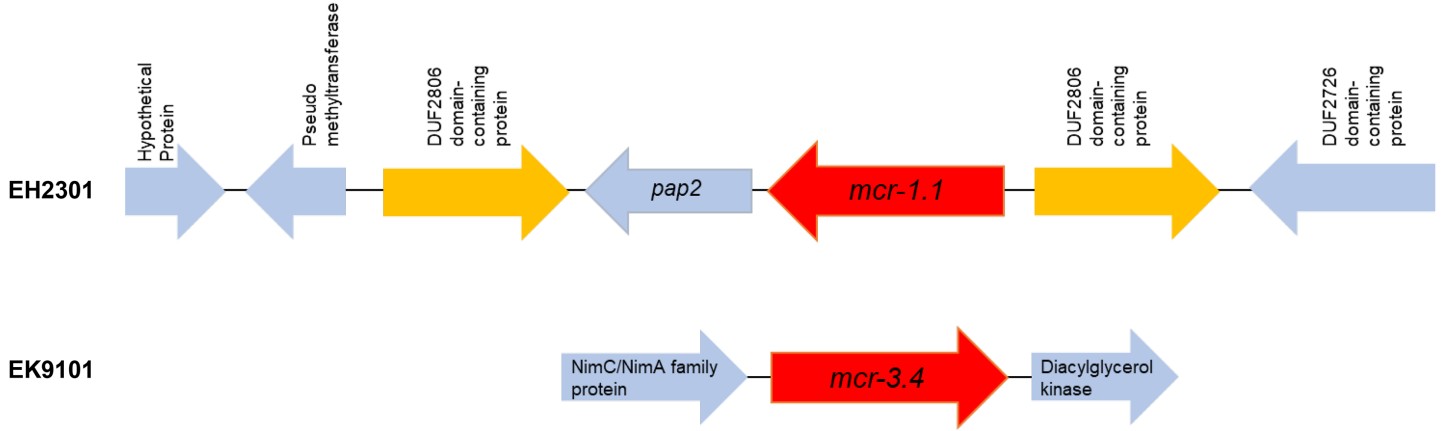

**Figure 5 Genomic content of EH2301 and EK9101 isolates carrying *mcr-1.1* and *mcr-3.4*, respectively.** The schematic shows the genes flanking the *mcr* genes in each isolate.

*1.1* gene, which the organization of the *mcr-3.4* gene in the EK9101 isolate was located between *nimC/nimA* and diacylglyceral kinase (*dgkA*) genes (Fig. 5).

As shown in Table 2, of the 10 isolates, two ESBL-producing *E. coli* carried $bla_{OXA-1}$. The EH2102 isolate carried $bla_{CTX-M15}$ and $bla_{OXA-1}$, whereas the EH9101 isolate contained $bla_{CTX-M15}$, $bla_{TEM-1}$, and $bla_{OXA-1}$. Nine different serotypes and eight sequence

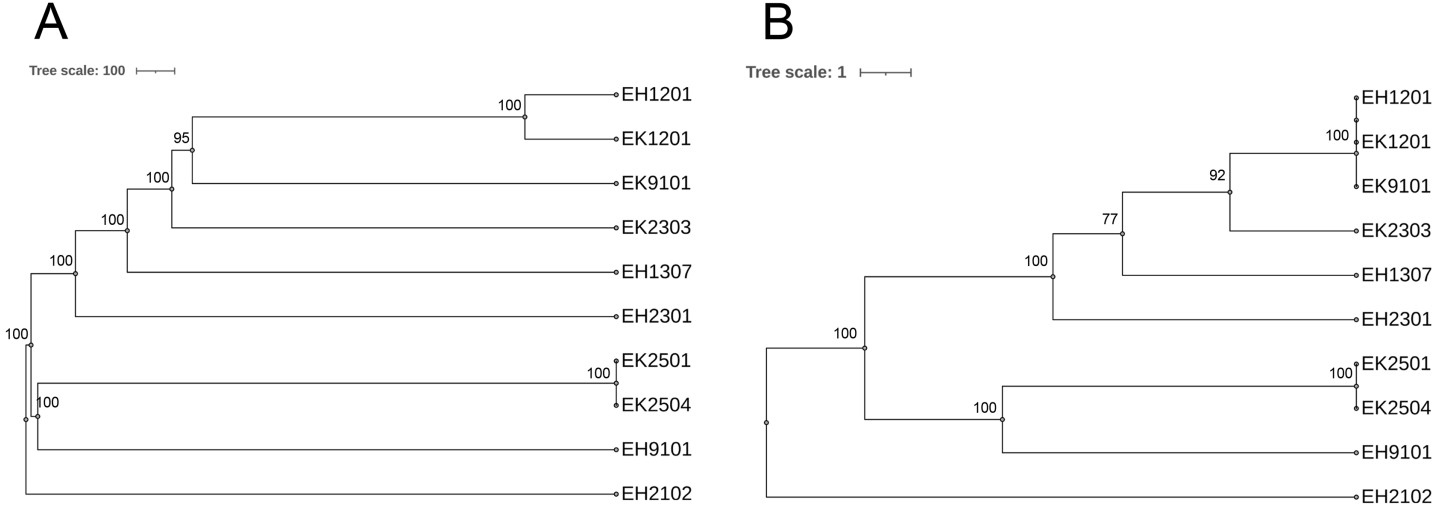

**Figure 6 Phylogenetic tree of selected 10 isolates based on (A) the whole genome MLST (wgMLST) and (B) the most discriminatory refinement loci (canonical wgMLST) using the web server, cano-wgMLST_BacCompare.** The numbers at the nodes represent the bootstrap percentages. Scale bars represent the branch lengths.

types (STs) were found. The pan-genome allele database of all 10 bacterial isolates was used to construct a wgMLST, using the cano-wgMLST_BacCompare analysis platform.

The PGAdb contained 9,715 genes, of which 3,171 (32.6%) genes belonged to the core genome, 3,044 (31.3%) belonged to the accessory genome, and 3,500 (36.0%) were unique genes. The phylogenetic relationships of the whole genomes and the identification of the most discriminatory loci are demonstrated in Fig. 6. The top 25 discriminatory refinement loci among the 10 *E. coli* genomes that used for constructing the canonical wgMLST tree are listed in Table S3. The isolates collected from different rivers at different time points were found to be closely related (ST224 in EK1201 and EH1201, and ST5218 in EK9101). The two isolates (EH1201 and EK1201) classified as ST224, however, carried different *bla* genes ($bla_{CTX-M14}$ and $bla_{CTX-M55}$, respectively). The EH1307 isolate, which is a novel ST (ST13160), is shown to be more related to EH1201 (ST224), EK1201 (ST224), EK9101 (5218), and EK2303 (ST648) when compared to other isolates.

## DISCUSSION

Typically, rice is planted and harvested from May to December in Thailand. The wet season runs from the middle of May to the middle of October and the dry season runs from mid of October to the early of May. Agricultural activities are based alongside the Kok River and Kham River in Chiang Rai, Thailand, where rice is mostly available. In-season rice farming continues from May to October. Usually, organic fertilizer and animal manure are applied before starting the rice planting in August and during the growth stage in July and September to improve the agricultural product and productivity. The levels of coliform bacteria were high in June and August in both the Kok River and Kham River, respectively, which was during the wet season in Thailand. Water flow during the monsoon may deliver soil and microorganisms to the river. The predominant agriculture during the wet season along both the Kok and Kham rivers was in-season rice, maize, and

cassava. Similarly, studies from the Chao Phraya River (central Thailand) demonstrated a strong trend of fecal-coliform concentrations during the wet season (*Huang et al., 2019*; *Singkran et al., 2018*). *Gao et al. (2015)* reported the transmission of antibiotic resistant bacteria from swine manure to the environment and rainfall and runoff were found to be associated with the spread of those bacteria to water (*Curriero et al., 2001*). Overall, total coliform bacteria collected at each site of the Kok River did not show any difference, except in January and February (dry season), in which sites A.3 and A.2 were found to have a higher number than that of site A.1, respectively. Higher coliform counts were also found in the Kham River in December, January, March, and April, both in sites B.2 and B.3, when compared to site B.1. In comparison to site B.1, the cumulative number of coliform bacteria at both sites during the dry season could be attributed to waste from urban and rural communities or agricultural processes along the river. Clearly, the evidence points to wastewater collected from the city as being the source of *E. coli* contamination distributed into the environment.

The occurrence of MDR *E. coli* was moderate in both rivers (45.3%). This circumstance may increase the incidence of transfer of resistance genes from non-clinical settings to a wide range of bacteria species in aquatic environments *via* horizontal gene transfer (*Taylor, Verner-Jeffreys & Baker-Austin, 2011*). In the present work, most *E. coli* stains were resistant to ampicillin and tetracycline. This is in agreement with a previous study, related to *E. coli* isolated from patients in a tertiary care hospital in Phayao, which is close to Chiang Rai province (*Srimora et al., 2021*). A study in ESBL-producing *E. coli* from vegetables in Chiang Rai demonstrated that most isolates were resistant to aztreonam, gentamicin and trimethoprim/sulfonamide (*Chotinantakul, Woottisin & Okada, 2022*), while ESBL isolates in this work were occasionally resistant to gentamicin and trimethoprim/sulfonamide. Taken together, our data would suggest that river waters are likely to be the reservoir of MDR *E. coli* that could disseminate resistance genes over extensive areas.

Most *E. coli* observed in this study belong to the phylogroup B1 (46.5%), A (17.4%), and C (16.3%), in accordance with a previous study (*Chotinantakul, Woottisin & Okada, 2022*). On the other hand, phylogroup A was the predominant type isolated from dairy farm wastewater in Chiang Mai (*Saekhow & Sriphannam, 2021*). Phylogroups A and B1 are ubiquitous in humans and animals, respectively (*Berthe et al., 2013*) and an infrequent phylogenetic group C is closely related to phylogroup B1 (*Moissenet et al., 2010*). Strains belonging to phylogroups B2, D, and F are related to extraintestinal *E. coli* infection (*Clermont et al., 2013*). The data here suggests that a high proportion of phylogroup B1 would be due to the contamination of organic manure that is commonly used in farming in this area. The use of organic manure, such as feces from chickens, pigs, and cows, would be the origin of the antibiotic-resistant bacteria because of the use of antimicrobials in livestock. Phylogroups A and C would possibly be from human contamination. Although some phylogroups are considered commensal, they could be converted to pathogens when receiving some antibiotic resistant determinants or virulence factor genes from the pathogenic ones.

The prevalence of ESBL-producing *E. coli* in this study was present at 22.1%. The isolation of ESBL-producing *E. coli* collected from water canal in central provinces of Thailand was 10% (*Boonyasiri et al., 2014*) and from wastewater sample in Chiang Mai, northern Thailand, were 88.7% (*Saekhow & Sriphannam, 2021*). All ESBL isolates in this work carried the $bla_{CTX}$ gene, with sporadic coexistence with the $bla_{TEM-1}$ gene, in accordance with a previous study (*Hassen et al., 2020*). However, the characterization of *E. coli* from dairy farm wastewater and pigs in northern Thailand demonstrated a higher rate of $bla_{CTX-M-positive}$ *E. coli* in combination with the $bla_{TEM-1}$ gene (*Lay et al., 2021*; *Saekhow & Sriphannam, 2021*). One ESBL-positive strain in this work contained both $bla_{CTX-M-55}$ and $bla_{TEM-116}$ genes. TEM-116 is thought to have evolved from TEM-1 (*Usha et al., 2008*). A study in Thailand reported the co-presence of $bla_{TEM-1}$ with $bla_{TEM-116}$ genes and $bla_{CTX-M-15}$ with $bla_{TEM-116}$ genes in clinical isolates of *E. coli* and *K. pneumoniae* (*Pornsinchai et al., 2015*), while another study reported the occurrence of the $bla_{TEM-116}$ gene from *E. coli* in poultry meat (*Tansawai et al., 2018*). There is no report of both $bla_{CTX-M-55}$ and $bla_{TEM-116}$ genes co-harboring in *E. coli* in Thailand, but it has been shown in piglets in Taiwan and environments in India (*Lee & Yeh, 2017*; *Murugadas et al., 2021*). TEM-116 may be transferred between intraspecies or interspecies *via* conjugation in the environment (*Lahlaoui et al., 2011*). Among ESBL-positive isolates in the present work, the $bla_{CTX-M-15}$ gene was predominant (44.7%), followed by $bla_{CTX-M-55}$ (26.3%), $bla_{CTX-M-14}$ (18.4%), and $bla_{CTX-M-27}$ (10.5%) genes. On the other hand, Runcharoen et al demonstrated a high prevalence of $bla_{CTX-M-55}$ followed by $bla_{CTX-M-14}$ and $bla_{CTX-M-15}$ genes in ESBL-producing *E. coli* isolated from farm waste and canals in eastern Thailand (*Runcharoen et al., 2017*). In northern Thailand, CTX-M-55 and CTX-M-14 were prevalent in ESBL isolates from healthy humans and pigs (*Lay et al., 2021*; *Seenama, Thamlikitkul & Ratthawongjirakul, 2019*). Furthermore, CTX-M-55 was found in ESBL-producing *E. coli* cultured from fresh vegetables (*Chotinantakul, Woottisin & Okada, 2022*). The occurrence of ESBL-positive strains emphasizes the importance of antimicrobial-resistant bacteria that can be distributed in the main rivers of Chiang Rai. Those rivers are used for consumption and farming activities, but by chance may increase the risk of MDR dissemination in humans, causing harmful diseases. Besides the finding in the river water, other environments such as soil, wastewater from hospitals and factories, and manure should be further monitored to explore the source of contamination.

Integrons play a key role in the dissemination and spread of antibiotic resistance by their ability to excise and integrate gene cassettes carrying antibiotic resistant genes (*Deng et al., 2015*). Integrons are widely spread in association with mobile DNA elements, *i.e.*, transposons or plasmids (*Deng et al., 2015*). A high proportion of ESBL-producing *E. coli* in the present work harbored the *int1* gene, in which some isolates were associated with either transposon Tn*3* or insertion sequence IS*Ecp1* genes on the plasmids. Two ESBL isolates harbored both class 1 and class 2 integrons. The presence of transposons and insertion sequences suggested the ability to mobilize many genes, particularly antibiotic resistance genes (*Razavi et al., 2020*). Previous work demonstrated the abundance of Tn*3* and IS*Ecp1* in most environments, *i.e.*, rivers, industrial pollutants, wastewater, marine, soil, and sediment (*Razavi et al., 2020*). The association of the $bla_{CTX-M-15}$ gene with an

upstream located IS*Ecp1* element was often reported (*Irrgang et al., 2017*; *Smet et al., 2010*). CTX-M-15-producing *E. coli* from German food samples were associated with plasmid and additional antimicrobial resistance genes as well as class 1 integron (*Irrgang et al., 2017*). Two dominant clonal lineages (ST167 and ST410) found in this study were also prevalent in samples of human and animal origin within the same sampling period. This data suggested the probable route of transmission of CTX-M-15-producing *E. coli* from livestock and food products, and consumers might be at risk from food contaminated with ESBL gene (*Irrgang et al., 2017*). Cumulatively, integrons and transposons that were simultaneously identified, together with *bla* genes in ESBL isolates, in this work could probably play a role in the dissemination of antibiotic resistance from contaminated water to human.

Our study revealed eight ESBL-producing STs from 10 selected isolates (ST69, ST131, ST224, ST603, ST648, ST1421, ST5218, and ST13160). ST131, ST69, and ST648 are the predominant extraintestinal pathogenic *E. coli* isolates worldwide (*Manges et al., 2019*). ST131, which carried ESBL genes, was found in the Thai patients and environmental isolates (*Runcharoen et al., 2017*). Previous studies reported that ESBL-producing *E. coli* belonging to ST224, ST603, ST1421, and ST5281 were occasionally found in humans and animals (*Apostolakos et al., 2017*; *Prapasawat et al., 2017*; *Qiu et al., 2019*; *Silva et al., 2016*). A newly identified ST in the present work was ST13160, which carried $bla_{\text{CTX-M-14}}$. The phylogenetic analysis did not show the unique genes found in each river or at any time point of collection. The ST224 and ST5218 isolates were found to be closely related, but the two ST224 isolates carried different *bla* genes. A novel ST13160 from EH1307 was shown to be more related to EH1201 (ST224), EK1201 (ST224), EK9101 (5218), and EK2303 (ST648) isolates. A limitation of this analysis is that only five ESBL isolates of each river were selected for sequencing. Analysis of more of the ESBL isolates could potentially enhance the relevance of this result.

The emergence of plasmid-mediated colistin resistance genes is of global concern. The distribution of *mcr-1* is more frequent than other types (*mcr-1* through *mcr-10*), particularly in food animals than in humans and food products, suggesting the role of foodborne transmission (*Elbediwi et al., 2019*). In this study, $bla_{\text{CTX-M55}}$ and $bla_{\text{TEM-1B}}$ were found with *mcr-1* in *E. coli*. The contig 45 harboring *mcr-1.1* of EH2301 showed 99% identity to plasmids pEC931, pCP52E-IncX4, and PN23, confirming that the *mcr-1.1* was located on IncX4 plasmid. IncX4, IncI2, and IncHI2 have been shown to be the predominant plasmid types carrying *mcr-1* spreading worldwide, including in Thailand (*Paveenkittiporn et al., 2021*; *Wu et al., 2018*). These plasmid replicons carrying the *mcr-1* gene could improve host fitness and co-selection, allowing *E. coli* to disseminate globally (*Wu et al., 2018*). A previous work demonstrated that *mcr-1.1* genes of colistin-resistant *E. coli* from swine were located mainly on IncHI2 and IncX4 types (*Garcia-Menino et al., 2019*). In Thailand, a study of plasmid replicon types, carrying *mcr-1.1* in clinical carbapenem-resistant Enterobacterales, revealed that IncX4 was the most common (*Paveenkittiporn et al., 2021*). Cumulatively, these data suggest the wide distribution of IncX4 plasmid harboring *mcr-1* among bacterial isolates of animal, environment, and human origins.

The genomic context analysis of the genome location of *mcr-1.1* revealed that of the *mcr-1.1-pap2* cassette was present in this work. A variety of *mcr-1.1-pap2* cassette compositions have been shown, suggesting the ability of *mcr-1* to mobilize across the genes (*Girardello et al., 2021*; *Snesrud et al., 2016*). IS*Apl1* flanking the *mcr-1.1-pap2* cassette conferred the transposition and was found to be lost after mobilization (*Snesrud et al., 2016*). However, this IS element was not observed in the present work. The disruption of the DUF2806-domain containing gene by the *mcr-1.1-pap2* identified in this work was previously described in chicken meat and slaughterhouse (accession numbers: MK875286.1; CP053735.1). The presence of a flanking DUF2726-domain containing gene upstream of the *mcr-1.1-pap2* cassette was discovered in clinically *mcr*-harboring carbapenem-resistant *E. coli* and *Klebsiella pneumoniae* isolates in Thailand (*Paveenkittiporn et al., 2021*). The MICs of colistin-resistant isolates were found near to be the resistance breakpoint (≥4 μg/ml), as previously described (*Lee et al., 2019*). Additionally, co-harboring of the *mcr-3.4* gene and the $bla_{CTX-M55}$ gene was found in the present work. The plasmid type carrying *mcr-3.4* could not be analyzed in this work due to the limited size of the contig. However, the identity of the contig when aligned with the reference genome on the NCBI database demonstrated the location on the plasmid pK18EC051 region (accession number CP049300). This location was shown to be on the region of *nimC/nimA-mcr-3.4-dgkA*. The *mcr-3.4* gene is a variant of *mcr-3.1* and was first reported in *E. coli* in China (*Xu et al., 2018*). The phenomenon of *mcr-3* gene distribution has been shown in several sources, including water, animals, food, and humans (*Elbediwi et al., 2019*). A previous study demonstrated the co-existence of ESBL ($bla_{CTX-M-14}$ and $bla_{CTX-M-55}$) and *mcr* genes (*mcr-1.1* and *mcr-3.1*) in pigs in Thailand (*Trongjit & Chuanchuen, 2021*). To our knowledge, the *mcr-3.4* variant has never been reported in Thailand, and our data revealed for the first time that the *mcr-3.4* gene co-occurred with the $bla_{CTX-M}$ gene in the river water.

## CONCLUSIONS

In conclusion, MDR *E. coli* was found in two main rivers, the Kok River and the Kham River, in Chiang Rai, Thailand. ESBL-producing *E. coli* was sporadically found, which mostly contained CTX-M-15. Co-occurrence of *mcr* genes (*mcr-1.1* and *mcr-3.4*) and ESBL genes were discovered, which were found in the river water. Integrons, transposons, and insertion sequences were also found in combination with the $bla_{CTX-M}$ genes, suggesting their role in disseminating the antibiotic resistant genes in the environment and possibly causing increasing risks to public health. Further findings of ESBL-producing *E. coli* should be extended to samples collected from soil and farmers near the rivers, including manures that are used in the agriculture. Wastewater from hospitals and small factories in the city should also be observed to find out the possibility of spreading those drug-resistant bacteria into the environment.

## ACKNOWLEDGEMENTS

We would like to thank Dr. Roger Callaghan for English-editing assistance.

### Funding

This work was supported by Mae Fah Luang University, Thailand (641C08004). The funders had no role in study design, data collection and analysis, decision to publish, or preparation of the manuscript.

### Grant Disclosures

The following grant information was disclosed by the authors:
Mae Fah Luang University, Thailand: 641C08004.

### Competing Interests

The authors declare that they have no competing interests.

### Author Contributions

- Kamonnaree Chotinantakul conceived and designed the experiments, performed the experiments, analyzed the data, prepared figures and/or tables, authored or reviewed drafts of the article, and approved the final draft.
- Pattranuch Chusri performed the experiments, analyzed the data, authored or reviewed drafts of the article, and approved the final draft.
- Seiji Okada conceived and designed the experiments, prepared figures and/or tables, authored or reviewed drafts of the article, and approved the final draft.

### Field Study Permissions

The following information was supplied relating to field study approvals (*i.e.*, approving body and any reference numbers):

Field experiments were approved by the Research Council of Mae Fah Luang University (project number: 641C08004).

### Data Availability

The data are available at the SRA: PRJNA846957.

### Supplemental Information

Supplemental information for this article can be found online at http://dx.doi.org/10.7717/peerj.14408#supplemental-information.

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
