# Peer review of "Detection and characterization of ESBL-producing Escherichia coli and additional co-existence with mcr genes from river water in northern Thailand"

_PeerJ, doi:10.7717/peerj.14408_

## Round 0.1 · original submission · Major Revisions

This study provided important information on the prevalence of antibiotic resistant strains in the Kok River and Kham River. The authors also assessed the distribution of coliform bacteria within a 10-month period; however, it has not been associated with the isolation of MDR or ESBL-producing E.coli.

It would be good if all the strains could be sequenced. If this cannot be done, please elaborate and explain the basis of selecting the 10 isolates for WGS.

Line 170: Has the expression study been done? Harboured the gene or expressed?

·

Basic reporting

Nothing major report. Italicize all scientific names.

Experimental design

whole-genome sequencing should be done in all isolates

Validity of the findings

By adding WGS for all isolates, the research will be more robust.

Additional comments

I believe the work is important and relevant, but its scope and validity will be enhanced by doing whole-genome sequencing for all isolates. Once that is done, a population structure analysis can be done along side the phylogenetic analysis. A population structure analysis will allow for the genetic architecture of the sampling to be examined with higher granularity. For that, I would use a hierarchical approach that encompasses genotyping the isolates at varying levels of resolution: BAPS1, ST, cgMLST, rMLST, wgMLST. On top of that, the authors can do a pan-genomic analysis with roary and identify accessory genome content that can represent distinct clusters in the sample. AMR, plasmids, all of those elements can also be mapped onto the population structure. Platforms such as Enterobase and ProkEvo should be helpful with the analysis. Thank you

·

Basic reporting

As an epidemiological study of ESBL-producing bacteria from rivers, the basic aspects of the study are considered to be in order. On the other hand, there are problems in the description of the discussion, and the conclusions drawn from the results do not follow a logical path in many discussions, requiring a complete revision. Simple mistakes are found here and there, and the English sentence structure is disorganized in some places.

Line 150
CFUs/100mL, isn't that a mistake for CFUs/mL? It would be better to bring this sentence after the ”... counting colonies.” on line 146?

Line 180-182
The English sentence structure is not ready.

Line 251
3.1% is a mistake for 31.

Line 253
The table number (Table 1) could be wrong.

Line 306
The figure number referred (Table S2) could be Fig.7.

Line 312-328
I do not know what impacts are expected from the agricultural season. Without information such as timing of fertilizer application, it is difficult to interpret. The last sentence of this paragraph describes rice cultivation, but of what importance?

Line 329-350
The first half of the paragraph is about drug resistance and the second half is about phylogenetic groups, which are not related. I don't see any evidence in context that phylogenetic group B1 is related to organic fertilizers. What exactly do you expect organic fertilizers to be used for? Is the use of organic fertilizers common in Thailand in the first place?

Line 351-382 This chapter contains three parts: 1. the incidence of ESBL E. coli; 2. coexistence with blaTEM; 3. the frequency of CTX-M subtypes. The frequency of appearance of CTX-M subtype is shown as data, but there is no mention of frequency of appearance in the conclusion.

Line 383-395
The first half describes transposons, but the conclusion changes to plasmid replicons. This should be corrected so that the logic can be connected.

Line 400-401 The description is as if the authors detected it from humans or animals; it should be changed to "It is reported that" or something similar.

Fig.1
The characteristics of the areas listed in Lines 127-136 should be included in Fig.1..

Fig.2
Is the temperature in Fig.2 air or water temperature?

Experimental design

Provide definition of “multi drug resistant”.

Line. 144
Please add the amount of specimen that was filtered.

Line 170,426
“Expression” was used for detected genes, but expression have not confirmed in this study. It must be changed to “possess” or “harbor”.

Line 186
What criteria were used to select the strain for WGS?

Line 192
Only MiSeq sequence data was used for assembly, so they are draft, not complete.

Validity of the findings

Line 249, 351
The prevalence of ESBLs is considered to be a percentage of colonies developed on the selective medium and not a percentage of E. coli detected. It seems to be difficult to compare to data reported by other investigators.

Line 306-307, 405
It is questionable to determine the same cluster based on cano-wgMLST results among ST types that even standard MLST cannot be determined to be clonal. cano-wgMLST results should be examined carefully.

Line 439-442
ESBLs are usually linked to IS and transposons. Clearly provide evidence that simultaneous detection is associated with increased risk.

Reviewer 3 ·

Basic reporting

The authors presented the study clearly with sufficient literature support. All relevant data are provided. There are only minor issues to be addressed:
Line 56: Please correct and standardize the protein family name throughout the manuscript, including Figure 6.
Line 137: Standardize the naming of the sampling sites.
Line 174: Please correct the sentence to “Three simplex PCRs were used to detect …”
Line 180: The term “single PCR” is confusing. Does it refer to a multiplex or a simplex PCR?
Line 180-182: The last two sentences of the paragraph are confusing. Please revise to clarify.
Line 186: I suggest the authors state the criteria or rationale for selecting these E. coli strains for WGS analyses.
Line 188-189: A table can be added to the results section (or as a supplementary table) to report the draft genomes features.
Line 205: Since only draft genomes were generated, I suggest changing the term “complete sequence” to “… draft genomes of all ten E. coli strains in this study …”.
Line 238: The findings of the colistin broth-microdilution are missing from this section.
Line 251: Should be 31%, not 3.1%.
Line 252-253: Table 1 does not reflect the AMR phenotypes of the ESBL-producing strains. An additional column can be added to include either the list of antimicrobial agents that the strains are resistant to or the arbitrarily assigned AMR pattern no. (shown in Table S1).
Line 258: Information in Figure 4 has already been stated in the text. The figure can be removed to avoid redundancy.
Line 269: The information in Figure 5 is redundant with the text.
Line 303-304: The meaning of this sentence is obscure. Please clarify what “the integration of the extraction of the whole genomes” means.
Figure 7: Please specify what the numbers at the nodes denote.

Experimental design

The experimental design is technically sound and methods are described in sufficient detail. There are only minor issues to address:
Line 188: Illumina MiSeq platform can generate paired-end reads with 300 bp. Is there a reason for the short reads generated in this study? Also, based on the Bioproject’s record in NCBI GenBank, Illumina HiSeq 2500 was used for generating the reads. Please clarify.
Line 274-275: Authors should state the rationale for selecting these ten strains out of the 38 ESBL-producing strains.
Line 290-291: I suggest the authors map the contig carrying the mcr-3.4 gene to a reference genome or perform a BLASTn analysis to determine if the gene is found on a chromosome or a plasmid region.

Validity of the findings

No comment.

Additional comments

No comment.

---

## Round 0.2 · Minor Revisions

Please amend the phylogenetic tree as described by Reviewer 3 and resubmit the manuscript.

·

Basic reporting

OK

Experimental design

The most crucial issue for me continues to be the lack of WGS for all genomes.

Validity of the findings

Needs WGS for all genomes.

Additional comments

I believe it is still important that all genomes be sequenced for this project.

·

Basic reporting

I have no comments.

Experimental design

I have no comments.

Validity of the findings

I have no comments.

Additional comments

I have no comments.

Reviewer 3 ·

Basic reporting

The authors have addressed all comments appropriately. There is just one technical glitch in Figure 6 (Phylogenetic tree). The bootstrap support in the tree should be changed to percentages (as stated in the figure legend) instead of decimals.

Experimental design

The authors have addressed all comments appropriately. Thank you.

Validity of the findings

No comment

Additional comments

No comment

---

## Round 0.3 · accepted · Accept

All the comments have been well-addressed.